# Osteolytic Bone Loss and Skeletal Deformities in a Mouse Model for Early-Onset Paget’s Disease of Bone with *PFN1* Mutation Are Treatable by Alendronate

**DOI:** 10.3390/ph16101395

**Published:** 2023-10-02

**Authors:** Zhu Ling, Hailati Aini, Shuhei Kajikawa, Jumpei Shirakawa, Kunikazu Tsuji, Yoshinori Asou, Hideyuki Koga, Ichiro Sekiya, Akira Nifuji, Masaki Noda, Yoichi Ezura

**Affiliations:** 1Department of Joint Surgery and Sports Medicine, Tokyo Medical and Dental University (TMDU), Tokyo 170-8455, Japan; zl8033833523@gmail.com (Z.L.);; 2Department of Nano-Bioscience, Tokyo Medical and Dental University (TMDU), Tokyo 113-8510, Japan; 3Department of Veterinary Medicine, Okayama University of Science, Imabari 794-8555, Japan; 4Department of Oral and Maxillofacial Surgery, Faculty of Medicine, University of Miyazaki, Kiyotake, Miyazaki 889-1692, Japan; 5Department of Orthopedic Surgery, Tokyo Medical and Dental University (TMDU), Tokyo 113-8510, Japan; 6Center for Stem Cell and Regenerative Medicine, Tokyo Medical and Dental University (TMDU), Tokyo 113-8510, Japan; 7Department of Pharmacology, Tsurumi University School of Dental Medicine, Tsurumi, Yokohama 230-8501, Japan; 8Faculty of Occupational Therapy, Teikyo Heisei University, Tokyo 170-8445, Japan

**Keywords:** alendronate, Paget’s disease of bone, profilin 1, osteoclast, bone deformity

## Abstract

A novel osteolytic disorder due to *PFN1* mutation was discovered recently as early-onset Paget’s disease of bone (PDB). Bone loss and pain in adult PDB patients have been treated using bisphosphonates. However, therapeutic strategies for this specific disorder have not been established. Here, we evaluated the efficiency of alendronate (ALN) on a mutant mouse line, recapitulating this disorder. Five-week-old conditional osteoclast-specific *Pfn1*-deficient mice (*Pfn1*-cKO^OCL^) and control littermates (33 females and 22 males) were injected with ALN (0.1 mg/kg) or vehicle twice weekly until 8 weeks of age. After euthanizing, bone histomorphometric parameters and skeletal deformities were analyzed using 3D μCT images and histological sections. Three weeks of ALN administration significantly improved bone mass at the distal femur, L3 vertebra, and nose in *Pfn1*-cKO^OCL^ mice. Histologically increased osteoclasts with expanded distribution in the distal femur were normalized in these mice. Geometric bone shape analysis revealed a partial recovery from the distal femur deformity. A therapeutic dose of ALN from 5 to 8 weeks of age significantly improved systemic bone loss in *Pfn1*-cKO^OCL^ mice and femoral bone deformity. Our study suggests that preventive treatment of bony deformity in early-onset PDB is feasible.

## 1. Introduction

Paget’s disease of bone (PDB) is one of the most well-known osteolytic disorders, with a spectrum of clinical and genetic manifestations [1,2,3]. During the past 30 years, treatment of PDB has been greatly improved with effective bisphosphonates (BPs) [2]. However, in contrast to the therapeutic effects on bone mass and pain, skeletal deformities are not easily treated and sometimes require surgery [3,4]. A recent clinical project planned by Ralston et al. concerns skeletal deformities in long-term outcomes of the zoledronate treatment for PDB patients with sequestosome 1 gene (*SQSTM1*) mutations [5]. However, since human studies take a long time to obtain practical answers to the raised questions, in vivo animal studies may help guide us in the direction of strategies.

PDB is a heterogeneous disorder. Such heterogeneity could be explained partially by multiple different causative or susceptibility genes, and environmental factors [1,2,3]. PDB with mutations in the most frequent causative gene *SQSTM1* tends to develop moderate skeletal symptoms in late adulthood [6]. In contrast, PDB and its related disorders with mutations other than *SQSTM1* tend to develop early and prominent bone deformities. For example, the heterozygous gain of function mutation in the TNF receptor family 11A gene (*TNFRSF11A*; also called RANK) can cause familial symptoms with distinct diagnoses of familial expansile osteolysis and so on [6]. Homozygous zinc finger 687 gene (*ZNF687*) mutations were found in 2016 in Italian hereditary cases with earlier and different clinical symptoms [1,7].

More recently, researchers identified the heterozygous loss of function (LOF) mutations in the human profilin 1 (*PFN1*) gene in some hereditary and sporadic cases of PDB with earlier onset and more prominent clinical symptoms [8,9,10]. The clinical examinations also indicated the increased prevalence of neoplastic degeneration in these cases, either through the existence of the cases associated with osteosarcoma [8] or the cases associated with giant-cell tumors, in addition to the symptoms consistent with PDB [10]. Because of such distinctive symptoms and etiology from typical PDB, this disorder was described as “early-onset polyostotic PDB” or “early-onset PDB” in the literature [8,9,10]. 

Profilin 1 protein is one of the basic molecules regulating the reorganization of actin filaments in the cells [11]. Its function has long been studied and understood to promote the elongation of barbed ends of actin filaments, typically at the periphery of cell boundaries [11,12]. However, as Squarez and Rotty et al. indicated in 2015, profilin 1 has suppressive effects on the elongation of branched actin filaments [13,14]. The branched actin filaments are responsible for structuring cellular processes named lamellipodia and podosome to provide the movements of osteoclasts. Thus, it was reasonable to hypothesize that the loss of function in the mouse profilin 1 gene (*Pfn1)* may lead to activating the locomotive function of the osteoclasts, in contrast to a reduced cellular migration in many other cells. 

Indeed, before the discovery of human early-onset PDB cases with *PFN1* mutations, we have shown that the homozygous conditional deficiency of the *Pfn1* specifically in osteoclasts in the mice (*Pfn1*-cKO^OCL^) resulted in PDB-like bone symptoms with postnatal osteolytic skeletal deformities associated with secondary bone sclerotic lesions and growth defects [15]. Heterozygous *Pfn1* deficiency in osteoclasts also resulted in a milder skeletal phenotype in our experiments, indicating the *Pfn1* deficiency only in osteoclasts mostly explains the pathogenesis of human PDB with heterozygous *PFN1* mutations in humans. 

The pathogenic background of early-onset PDB with *PFN1* mutation is different from typical PDB or related diseases. The enhanced potential in osteoclastogenesis in culture is common in typical PDB. The skeletal lesions of PDB patients often include giant multinucleated osteoclasts in histology. However, in our mutant line, such findings were not obvious [15]. Instead, from the aspect of actin filament reorganization and cellular movements, we have shown that our mice had osteoclasts with increased migratory activity, increased cellular processes, and increased bone resorption activity, which were mutually correlated in culture experiments [15,16]. Thus, the distinct etiological background and clinical symptoms of the early-onset PDB with *PFN1* mutation from other PDB in humans may indicate a need for considering the different treatment plans for this type. In fact, treatment strategies for early-onset PDB with *PFN1* mutations have not been established yet. 

Therefore, to evaluate the therapeutic effect of the bisphosphonates in this disorder based on *PFN1* mutations, we utilized our mutant mouse line, *Pfn1*-cKO^OCL^. The effectiveness of the amino bisphosphonates including zoledronate and alendronate (ALN) is well-known for typical PDB, and zoledronate may be more effective than alendronate. However, since we were unfamiliar with using zoledronate in mouse experiments, we decided to administer the ALN to juvenile mice from 5 to 8 weeks of age, emphasizing the therapeutic effect on bony deformity and bone growth. This timing was chosen because the skeletal deformity became evident in just around 4–5 weeks [15]. A short term experiment was chosen because this was the first trial. 

## 2. Results

### 2.1. Administration of ALN Improved the Decreased Trabecular Bone Mass in the Distal Femurs of Pfn1-cKO^OCL^ Mice

We administered ALN to *Pfn1*-cKO^OCL^ and the control *Pfn1*^flox/flox^ littermates (we call them wild-type (WT) littermates hereafter) at 5–8 weeks of age (Figure 1A). Females (total n = 33) and males (total n = 22) were separately analyzed. However, since the outcome information was similar in both genders, we mainly describe the results of females here. Body weight gain in each mouse was overall constant during the experimental period of the 3 weeks of ALN administration (Figure 1A). 

Then we analyzed if the 3 weeks of ALN can prevent osteolytic bone loss at the distal femur in the *Pfn1*-cKO^OCL^ mice (Figure 1B). Without treatment, the quantitative bone mass evaluated by trabecular bone mass (BV/TV; %) was significantly smaller in *Pfn1*-cKO^OCL^ mice at about 23% than that of WT controls (Figure 1C). However, the ALN treatment significantly increased it by about 10.1 fold in *Pfn1*-cKO^OCL^ mice, in contrast to about 2.4-fold increment in WT mice (two-way ANOVA, Vehicle vs. ALN; *p* = 4.8 × 10^−^^11^, Figure 1C). Although the interaction between the genotypes and treatments was not evident using two-way ANOVA (Interactive *p*-value: 0.50), the therapeutic ratio of ALN to improve bone mass was significantly greater in the osteolytic *Pfn1*-cKO^OCL^ mice than in the WT (Figure 1D: *p* = 1.2 × 10^−^^4^, Student’s *t*-test). Other morphometric parameters of trabecular bone also showed reconcilable results (Table 1); similar results were confirmed in males (Appendix A). 

### 2.2. ALN Efficiently Suppressed the Increased Osteoclast Number in the Trabecular Bone Area of the Long Bones in Pfn1-cKO^OCL^ Mice

We next evaluated if the ALN successfully suppresses the increased number of osteoclasts in histological sections of distal femurs in *Pfn1*-KO^OCL^ mice by staining for tartrate-resistant acid phosphatase (TRAP) (Figure 2A). The number of osteoclasts per bone surface (Oc.N/BS) in *Pfn1*-cKO^OCL^ mice was about 2.5 fold greater than that of WT, while the parameter was suppressed by ALN treatment to about 25% in WT and about 22% in *Pfn1*-cKO^OCL^ mice (Figure 2B). Similarly, the osteoclast surface per bone surface (Oc.S/BS) was originally greater in Pfn1-cKO^OCL^ mice by about 2.6 fold, while it was suppressed by ALN to about 44% in WT and about 14% in *Pfn1*-cKO^OCL^ mice (Figure 2C). Thus, a significant interactive effect between the genotypes and ALN treatment was indicated by two-way ANOVA (Figure 2C). Similar findings were observed in proximal tibiae (Appendix A) as well as that in male femurs (data not shown). We also observed a significantly increased number of osteoclasts and their expanded distribution at the endosteal metaphyseal femoral cortex in *Pfn1*-cKO^OCL^ mice than in WT. The ALN suppressed them in both genotypes, and the difference was more prominent in the *Pfn1*-cKO^OCL^ mice (Figure 2D).

### 2.3. ALN Administration Partially Prevented the Progression of Long Bone Deformities in Pfn1-cKO^OCL^ Mice

The above-described remarkable changes in the distribution and the number of osteoclasts in the distal femur made us ask if the ALN also prevents the metaphyseal bone collar expansion in *Pfn1*-cKO^OCL^ mice. The shape of the distal femurs was geometrically quantified using 3D μCT images, with consideration of the proportional difference of the femoral length, which was shorter in *Pfn1*-cKO^OCL^ mice than that in WT littermates at about 90% level (*p* = 6.1 × 10^−^^6^; Appendix A and Figure 3A upper panel for females, and Appendix A for males). 

First, the cross-sectional images at midshaft (Figure 3A bottom) indicated a significantly greater cortical perimeter size with normalization of the *Pfn1*-cKO^OCL^ mice than that of WT littermates at about 120% level (Figure 3B), and the three weeks of ALN decrease it in females (*p* < 0.05 by two-way ANOVA). The difference regarding the suppression levels between the WT (at about 5%) and the *Pfn1*-cKO^OCL^ (at about 14%) was significant (Figure 3C; Student’s *t*-test). Although such reduction was not exactly proven in males possibly because of its smaller sample size (Appendix A), this reduction was regarded as therapeutic in the mutant mice. 

The characteristic difference of the contour curvature between the genotypes was noticeable especially from the midshaft to the distal metaphysis. In WT, this part was basically concave (indicated by red curves in Figure 4A), whereas in *Pfn1*-cKO^OCL^ mice it was convex (indicated by blue curves in Figure 4A). Since it was evident that the ratio of the curve length in convex versus concave was significantly different between the genotypes (Appendix A), we analyzed the effect of ALN on curvature change separately by genotype. In WT, the concave curve from the midshaft to the distal quartile was analyzed (Figure 4B), whereas the convex curve from the midshaft to the epiphyseal growth plate was analyzed in *Pfn1*-cKO^OCL^ mice (Figure 4C). Then, it was evidenced that ALN treatment significantly suppressed the convex curvature in *Pfn1*-cKO^OCL^ mice but not the concave curvature in WT femurs.

This therapeutic effect of ALN for the femoral shape was also indicated by the changes in the mediolateral (ML) broadness of the femur at the third quartile (Figure 4A, horizontal yellow arrows at DQ); it was increased in WT but was decreased in *Pfn1*-cKO^OCL^ mice. Indeed, the broadness normalized by that of the distal end was significantly greater in the Pfn1-cKO^OCL^ mouse femurs (*p* = 1.0 × 10^−^^7^) than the WT (Figure 4D) and was oppositely influenced by the ALN treatment in both genotypes; there was a significant interaction between genotype and ALN administration (*p* = 0.01). Such a difference was also indicated by the therapeutic ratio of this index (*p* = 0.0027; Student’s *t*-test, Figure 4E). Similar results were obtained for males (Appendix A). Therefore, these results indicate that only 3 weeks of ALN can prove the partial correction of the long bone deformity in *Pfn1*-cKO^OCL^ mice if a suitable geometric parameter was evaluated. 

### 2.4. ALN Administration Improved the Decreased Trabecular Bone Mass in the L3 Vertebra and Nasal Bone of Pfn1-cKO^OCL^ Mice but Not for Its Deformity

To examine if the above-described effect of ALN on the femurs can be observed in different sites of the skeleton, we first compared the effects of ALN in the lumbar vertebra. The trabecular bone mass analysis of the third vertebral body (L3) indicated a significant increase in BV/TV by ALN (Figure 5A,B), and the extent of the increment was significantly greater in *Pfn1*-cKO^OCL^ mice than in WT (Figure 5B,C). Other morphometric parameters of trabecular bone also showed reconcilable improvements (Table 2), and we observed similar results in males (Appendix A). 

Also, the effect of ALN on facial bone was evaluated in *Pfn1*-cKO^OCL^ mice. To this end, we focused on the nasal bone on the μCT images in the midsagittal plane and compared its BMC (mg). We haven’t chosen to compare the volumetric BMD (g/cm^3^) for our evaluation, because the *Pfn1*-cKO^OCL^ mice had apparently smaller nasal bones compared to the WT [15]; this was supposed to be the result of fragile bone fracture caused by daily mechanical stress. Thus, we compared the nasal bone BMC and showed that in the *Pfn1*-cKO^OCL^ mice it was significantly improved by ALN treatment (Figure 6A,B). Together, the systemic bone loss in the *Pfn1*-cKO^OCL^ mice was significantly improved by three weeks of ALN at the therapeutic dose from 5–8 weeks of age.

However, when we tried to detect the therapeutic effect of ALN on the shortening deformity of the L3 vertebra or the nasal bone; significant differences were not proven in such parameters (Appendix A). The relevance of the findings is to be discussed in the next section. Nevertheless, we conclude here, that only 3 weeks of ALN treatment in our mutant *Pfn1*-cKO^OCL^ mice resulted in a slight improvement in the skeletal deformities, at least in the distal femur. 

## 3. Discussion

A novel osteolytic disorder due to *PFN1* mutation was discovered recently as early-onset PDB [8,9]. In this study, we evaluated the effect of a therapeutic dose of ALN on skeletal deformities in *Pfn1*-cKO^OCL^ mice at 5–8 weeks of age, as a model for this novel disorder. As expected from clinical situations, the 3 weeks of ALN clearly improved bone loss significantly. In addition, we could show that skeletal phenotype regarding the shortening and ballooning of long bones was partially prevented. Although it was not easy to show a similar effect in vertebrae and in the nasal bone, our findings may suggest that further modification of therapeutic protocols may allow us to propose better preventive therapy for skeletal growth and deformity. 

The pathological basis of the novel early-onset PDB with *PFN1* mutations has not been clearly understood. However, since the skeletal symptoms of PDB are believed to be caused mainly by the dysregulated increase of osteoclast activity and their number, in general, it is reasonable to regard the pathological basis of the novel early-onset PDB with *PFN1* mutations could also be the dysregulated osteoclast function. Since we have already shown that the conditional *Pfn1* deficiency in osteoclasts exactly results in osteolytic bone loss [15], their symptoms should primarily depend on the osteoclast function. Indeed, in culture experiments, the increased osteoclast movements were associated with increased bone resorption activity [16]. However, since we have not exactly shown in our previous studies that increased osteoclast function and their number was the cause of the skeletal phenotype in our mutant line, this is the first time that we show the increased osteoclast function was the real cause of skeletal phenotype in our mutant mouse line; the extreme bone loss was almost eliminated by administration of short term subcutaneous ALN. 

The notion was also supported by histological findings based on TRAP staining. We showed that the ALN treatment not only suppressed the number of TRAP-positive osteoclasts but also improved their abnormal distribution pattern, especially at the endosteum. It is not surprising to see the osteoclast number decrease by ALN that directly targets the bone-resorbing osteoclasts. However, it was not sure if the altered distribution pattern of the endosteal osteoclasts could also be improved since there has not been shown how the increased osteoclast movements contribute to the altered distribution. The altered distribution was associated with the increased number of osteoclasts in vivo, whereas the osteoclast number was not increased significantly in culture experiments [15]. Therefore, we assume that increased movements of osteoclasts in vivo have some influence on osteoclast differentiation or its survival in vivo, which are possible targets of ALN [17,18]. Further investigations are necessary in the future.

The second question of the present study was about skeletal deformities. Since the skeletal deformities of the *Pfn1*-cKO^OCL^ mice were thought to depend on osteolytic bone loss [15], they should also be treated by inhibiting the osteoclast function. However, it has not been clearly shown by in vivo evidence. In the present study, this question was answered by the geometrical analysis of the 3D images in distal femurs at least in females. In addition to the curvature analysis using FIJI software (ImageJ2 version 1.0, Plugin “Kappa” version 2.0.0), relative length analysis of the cortical perimeter at midshaft cross-sections, as well as the distal quartile broadness in AP projection images created by 3D μCT effectively indicated that there was a significant therapeutic effect in the *Pfn1*-cKO^OCL^ mice when normalized with the lateral width of the distal femora. An interesting point to this observation was that ALN increased the broadness of the distal quartile of the femur, whereas it decreased the broadness in the *Pfn1*-cKO^OCL^ mice oppositely. The following are the thinkable interpretations of the observation: in WT, suppression of bone turnover by ALN possibly from the periosteum resulted in increased distal femur width, whereas in *Pfn1*-cKO^OCL^ mice ALN prevented the enhanced expansion of the width by suppressing the already increased bone resorption from the endosteum. This contrast may correspond to the pattern of the osteoclast distribution in the metaphysis of long bones as indicated in our previous [11] and present study. It should be noted that the affected bone turnover at this anatomical site in both ways is known to cause the macroscopic bone deformity named Erlenmeyer flask deformity of the metaphysis. The inhibition as well as enhancement of the bone turnover cause this deformity. 

The reasons why only a marginal improvement was observed might be explained in multiple ways. The intervention period of three weeks may have been too short. The timing of the intervention might be too late. In addition, a possible confounding factor was the adverse effect of bisphosphonates on the young growing bone. Caution must be paid regarding their dosage because an overdose of bisphosphonates results in disturbed longitudinal growth of long bones with characteristic Erlenmeyer flask deformity at least in mice [19], and maybe in humans [20]. Indeed, our experiments showed a slight influence on the length of long bones and vertebrae by ALN, whereas the difference was statistically negligible.

The reason why the therapeutic effect was limited only to the distal femur might be explained by the fact that the ectopic Cre expression was also limited only in the epiphyseal parts of the long bones. The loss of *Pfn1* in the mesenchymal progenitor cells in femoral metaphysis may suppress their movements, and it may result in reduced bone formation contributing to the bone widening only in the metaphysis of the long bones. In this explanation, it should be noted that the ectopic expression of Cre at this site [21] is inherently protective of deformity and does not act as a cause of the deformity. In the other sites of mutants’ bone, increased periosteal bone formation in response to the enhanced bone resorption from the inside may accelerate the deformity and longitudinal shortening, resulting in the inability to be treated by only three weeks of ALN in our experiment. An alternative likely explanation for the difference between the femur and other sites might be the difficulty in assessing morphological and geometric changes in the vertebrae and nasal bones. Therefore, the geometric assessment of deformities in these sites should be improved in future studies. 

Finally, to treat skeletal deformities in vivo, a prophylactic approach may have to be considered before destructive changes occur. Although we may have to add some arrangements for administrating the drug to the mice before weaning, we may consider it in future studies. The therapeutic window may have to be longer. Also, an important anatomical target site may not be the extremities but the facial bones, which may bring about a life-threatening condition. In this study, we chose the nasal bone because its phenotype is evident in mouse models and quantitative evaluation is easy. In actual clinical practice, however, mandibular morphology and alveolar bone changes are more important than nasal bone shape. Therefore, future measurements of the alveolar, mandibular, and maxillary bones should be made from this perspective and examined in more detail to evaluate the clinical usefulness of ALN treatment. Also, the observation of the mild adverse effects of ALN on bone growth indicated that we may have to consider arranging methods of drug administration, selecting the therapeutic agents including other amino bisphosphonates and other available agents, and may need to develop better therapeutic agents whose effects can be modulated controllably at an appropriate level. Our experimental system may be useful for developing such novel agents.

In summary, this study indicated that standard bisphosphonate therapy can treat bone loss in the early onset PDB based on *Pfn1* mutation in a mouse model system. In addition, our study also indicated that the skeletal deformity might be prevented or recovered by ALN at least partially even after disease onset. The limitations of our study were that the therapeutic effects of ALN on bony deformity were overall marginal and that different administration protocols were not tested in this study. Further investigations applying different administration protocols and therapeutic agents should be conducted in the future which may lead to establishing a better sophisticated therapeutic protocol to treat young patients with early onset PDB with skeletal deformities. Biologically, our results support the hypothesis that the bone phenotype of *Pfn1*-cKO^OCL^ mice is caused mainly by osteoclast-specific dysfunction. However, still, there is still a possibility that *Pfn1* mutation in mesenchymal progenitor cells in the cambium layer that gives rise to bone collar osteoblasts or chondrocytes may oppositely function to resist deformity.

## 4. Materials and Methods

### 4.1. Animals

The osteoclast-specific *Pfn1*-knockout (KO) mice (*Pfn1*-cKO^OCL^) were generated previously in our laboratory [15]. Briefly, the *Pfn1*^flox/flox^ mice were crossed with *Ctsk*-Cre knocked-in mice to obtain *Pfn1*-cKO^OCL^. Adolescents were genotyped by genomic PCR at postnatal 3 weeks. The experiment began at 5 weeks of age when the skeletal symptoms began to be evident. This timing was selected also because of its practical easiness to begin with. The ALN (0.01 mg/mL) was administered subcutaneously to these mice at the dose level of 0.1 mg/kg body weight, twice a week for 3 weeks in the ALN group, whereas only the vehicle was injected for the control mice (Figure 1A). At 8 weeks of age, the femur, tibia, skull, and the third lumbar vertebra were dissected to be used for morphological analysis (Figure 1A). To monitor the mouse health status, body weight was measured twice a week during the administration period (Figure 1B). All the experiments were approved by the animal welfare committee of Tokyo Medical and Dental University (Approval Number: A2017-082A, A2018-180A, and A2019-222A).

### 4.2. 3D-μCT Analysis for Bone Mass

Three-dimensional µCT images were obtained using the Scan-Xmate-E090 (Comscan Tecno, Sagamihara, Japan). Standard parameters for bone histomorphometry for trabecular bone, such as bone volume/tissue volume (BV/TV), trabecular number (Tb.N), trabecular thickness (Tb.Th), trabecular spacing (Tb.Spac), and trabecular separation (Tb.Sp) [22] were analyzed on distal femurs, using Tri/3D-Bon-FCS64 software (Ratoc System Engineering, Tokyo, Japan). The region of interest (ROI) in the distal femur was set from the level of 200 μm proximal to the anterior edge of the growth plate, and the thickness was set as 1000 μm (Figure 1B). Similarly, we evaluated the cortical bone parameters including cortical thickness and cortical perimeter length in the cortical bone area at femoral midshaft with 200 μm thickness (Figure 3A–C). For the third lumbar vertebral body (L3), the trabecular bone area with 500 μm thickness at the caudal part was analyzed (Figure 5A–D). In the facial bone, the area of the entire nasal bone was selected as ROI to evaluate the bone mineral content (BMC). 

### 4.3. Quantitative Analysis of Bone Deformity

We used 3D-μCT images to quantitatively evaluate the long bone and skull deformity. The cortical bone parameters including cortical thickness (Co.Th) and cortical perimeter (Co.Peri) were analyzed by cross-sectional images of the femoral midshaft, and these values were normalized against the width of the distal epiphyseal growth plate (W); the ratio of Co.Peri/W was compared among groups. To evaluate the size and shape of the whole femur, the anteroposterior (AP) projection of the 3D reconstructed μCT images was used. The characteristic Erlenmeyer deformity was evaluated by the contour curves of the AP projection (see Figure 4A). The curvature of the characteristic fractions around the midshaft to around the third quartile of the femoral contour was specifically measured, using Image J and FIJI software [23] with plugs in software named Kappa-curvature analysis. The measurement was performed by repeating ten times in each sample and the average value was used for the analysis. The medial and lateral parts of the contour curves were analyzed separately but merged by calculating the average in each bone. We also evaluated the normalized mediolateral broadness of distal metaphysis at the third quartile (DQ/W). 

### 4.4. Bone Resorptive Parameters by TRAP Staining in Bone Histomorphometry

For histological analysis, paraffin-embedded decalcified sections for the femurs and tibias were stained with a substrate for tartrate-resistant acid phosphatase (TRAP), according to a standard protocol as described [15]. Bone resorption parameters including osteoclast number/bone surface (Oc.N/BS) were analyzed based on the standard guideline ([24]; ASBMR). 

### 4.5. Statistical Analysis

Statistical analysis was performed on the data obtained by biological, histological, radiological, and physical measurements. When applicable, the Kolmogorov-Smirnov (K-S) test confirmed normal distribution. The data were presented by mean ± standard deviation (SD). Two-way analysis of variance (two-way ANOVA) was applied to evaluate the therapeutic and interactive effects of *Pfn1* mutation and ALN treatment. In addition, a Student’s *t*-test was applied to compare the differences between the genotypes in the therapeutic effect of ALN on the morphological changes in femurs, spines, and nasal bones. We regarded the differences as significant when the *p*-values were less than 5% (*; *p* < 0.05).

## 5. Conclusions

Our study indicated that skeletal phenotypes of the *Pfn1*-cKO^OCL^ mice including the bony deformity were improved by short-term ALN administration from 5 to 8 weeks of the mice.

## Figures and Tables

**Figure 1 pharmaceuticals-16-01395-f001:**
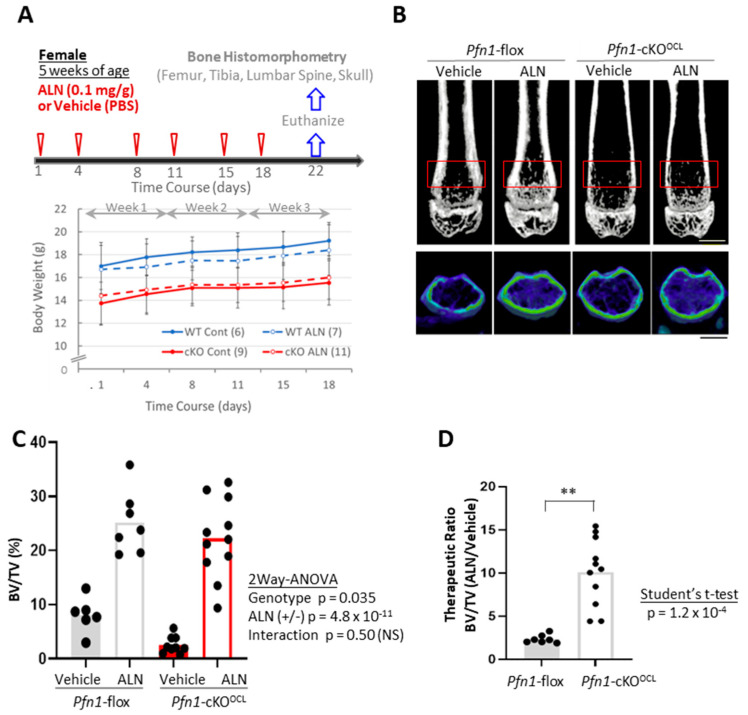
Bone mass analysis of *Pfn1*-cKO^OCL^ mouse femurs. (**A**). Schematical diagram of the therapeutic protocol by ALN. Chronological changes in body weight gain were plotted on the graph (bottom). A minimal trend of suppression by ALN for the body weight gain in WT littermates (blue lines) was statistically negligible. (**B**). Representative 2D (**upper**) and 3D (**bottom**) μCT images of the distal femurs in each experimental group are shown. The red boxes in the upper panel indicated the region of interest (ROI) for the bone mass analysis. Scale bar: 1 mm. (**C**). Trabecular bone mass (BV/TV: %) at distal femur metaphysis was analyzed using 3D μCT (**B**). (**D**). The relative effect of ALN treatment in the BV/TV in (**C**) was calculated by dividing the data in the mice treated with ALN with the mean values obtained from the mice treated only with the vehicle. Statistical analysis was performed using Student’s *t*-test. (**; *p* < 0.01), *p*; *p*-value.

**Figure 2 pharmaceuticals-16-01395-f002:**
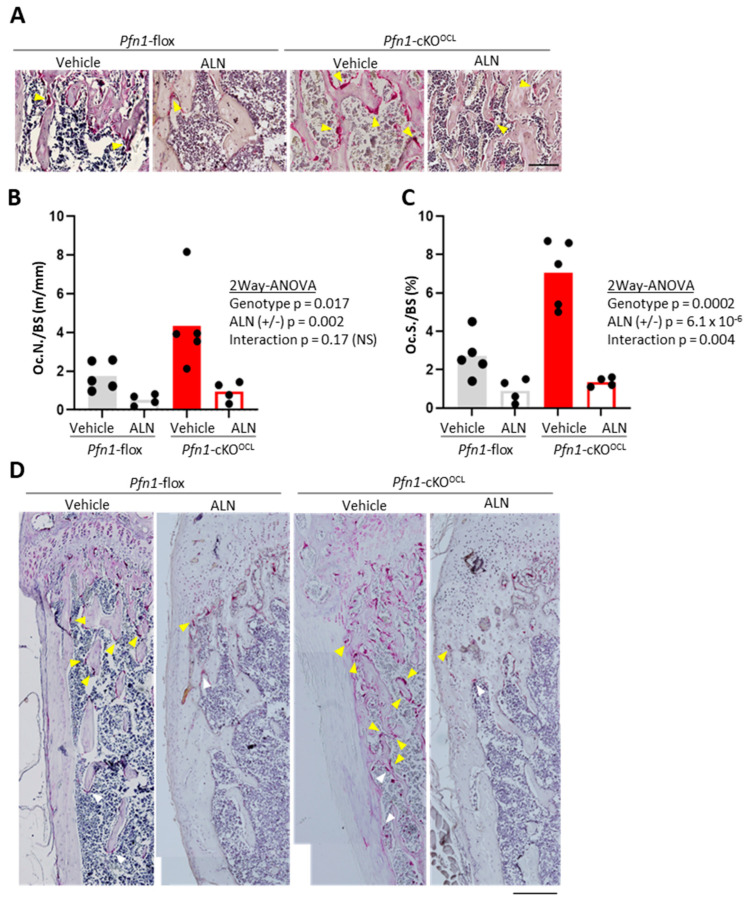
Bone resorptive parameters for distal femurs by TRAP staining. (**A**). Representative microscopic images for the secondary trabeculae in distal femurs. Multi-nucleated red cells with more than two nuclei (yellow arrowheads) were regarded as TRAP-positive osteoclasts. Scale bar: 100 μm (**B**). The histomorphometric bone resorption parameter, Oc.N/BS was plotted on the graphs and was statistically analyzed by comparing the experimental groups. (**C**). Oc.S/BS was plotted on the graphs and was statistically analyzed by comparing the experimental groups. (**D**). Representative histological images of TRAP staining in epiphyseal to the metaphyseal femoral cortex of the WT and *Pfn1*-cKO^OCL^ mice, indicating the distribution of the TRAP-positive cells (arrowheads; yellow multinucleated and white mononucleated cells). Scale bar: 200 μm.

**Figure 3 pharmaceuticals-16-01395-f003:**
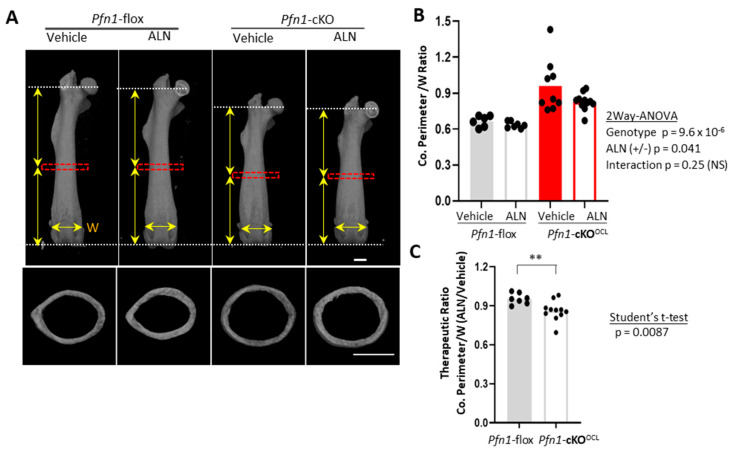
Morphometric analysis of *Pfn1*-cKO^OCL^ mouse femurs at midshaft. (**A**). Representative μCT images of the anteroposterior (AP) projection for entire femurs (**upper** panels) and cross-sectional images at midshaft (**bottom**) that were analyzed for cortical bone parameters in (**B**,**C**). Red boxes indicate the ROI for cortical bone analysis. Vertical arrows indicate the length of the proximal and distal halves. Horizontal arrows indicate the width (W) of the border between the distal metaphysis and the epiphyses at the growth plate level. Scale bar: 1 mm (**B**). Cortical Perimeter (Ct.Pm) normalized by mediolateral (ML) width of the epiphyseal growth plate (W) were plotted on the graph. Statistical analysis was performed using two-way ANOVA. (**C**). The therapeutic ratio of the above index by ALN was plotted on the graph to compare it between the WT and *Pfn1*-cKO^OCL^ mice by Student’s *t*-test (**; *p* < 0.01).

**Figure 4 pharmaceuticals-16-01395-f004:**
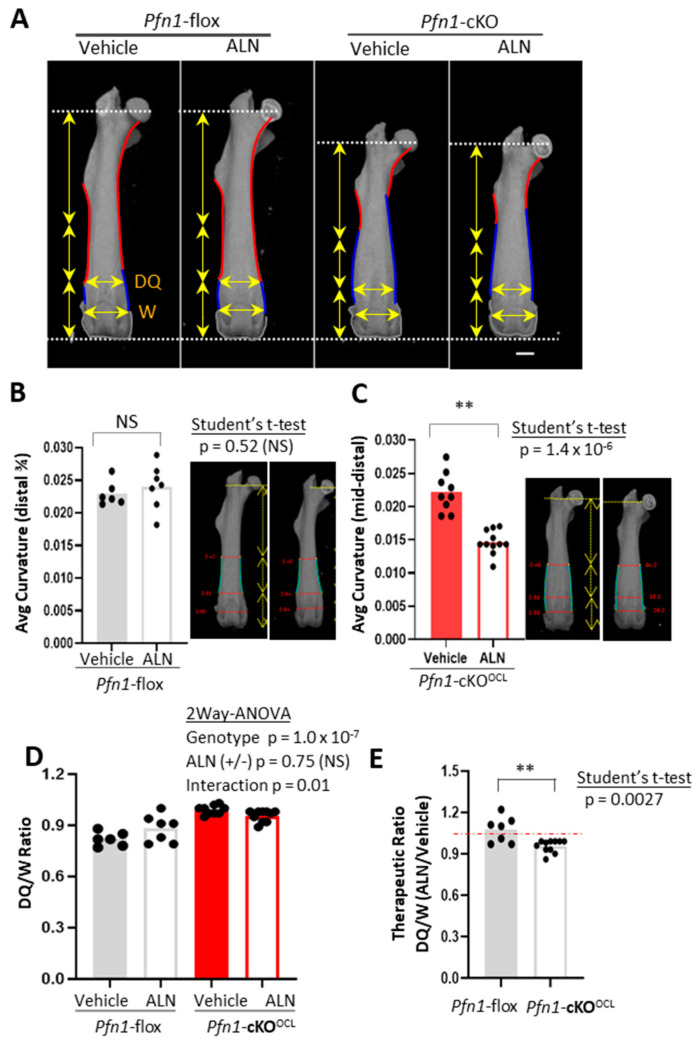
Geometric analysis of *Pfn1*-cKO^OCL^ femurs from midshaft to the distal metaphysis. (**A**). The outline of the AP projection μCT images represented in Figure 3A was traced with red and blue curves, indicating concave and convex features, respectively, in diaphysis to the metaphysis. The distal ends of the femurs are also traced with gray curves. Vertical arrows indicate the length of the proximal half, third, and fourth quartile. Horizontal arrows indicate the ML width of the distal quartile (DQ), and the distal growth plates (W). Scale bar: 1 mm (**B**). The average curvature in concave curves from midshaft to the distal quartile was compared between the control versus ALN-treated WT mice. representative image was shown on the right side. (**C**). The average curvature in distal convex curves from the midshaft to the distal growth plate was compared between the control versus ALN-treated *Pfn1*-cKO^OCL^ mice. A representative image was shown on the right side. (**D**). The distal quartile expansion parameter was calculated as the DQ/W ratio and analyzed by plotting and comparing among the groups. (**E**). The relative effect of ALN treatment in the DQ/W in panel D was statistically analyzed by Student’s *t*-test. (**; *p* < 0.01), NS; not significant.

**Figure 5 pharmaceuticals-16-01395-f005:**
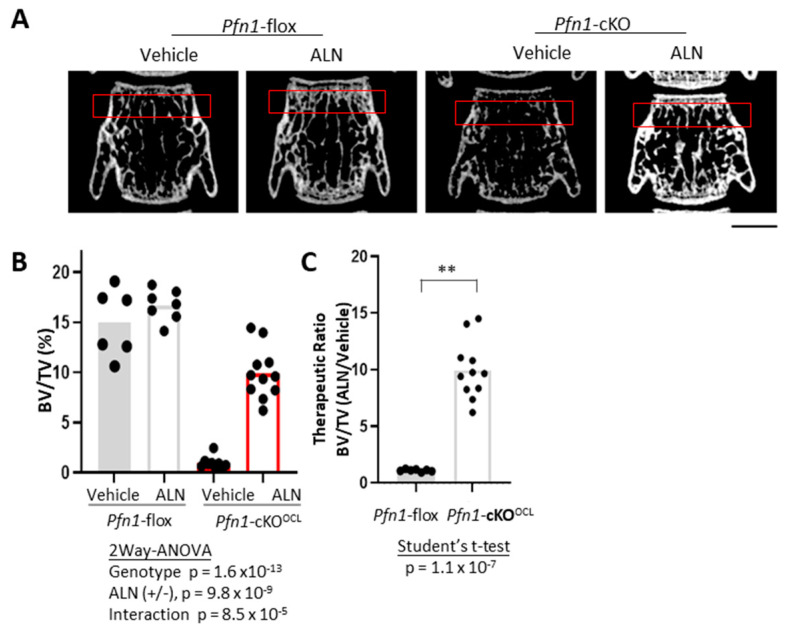
Bone mass analysis of *Pfn1*-cKO^OCL^ L3 vertebrae. (**A**). Representative 2D μCT images of frontal sections of the L3 vertebral body. Red boxes indicate the ROI for bone mass analysis. Scale bar: 1 mm (**B**). Trabecular bone mass at distal femur metaphysis was analyzed by BV/TV (%) using 3D μCT. (**C**). The relative effect of ALN treatment in the BV/TV in panel B was statistically analyzed by Student’s *t*-test. (**; *p* < 0.01).

**Figure 6 pharmaceuticals-16-01395-f006:**
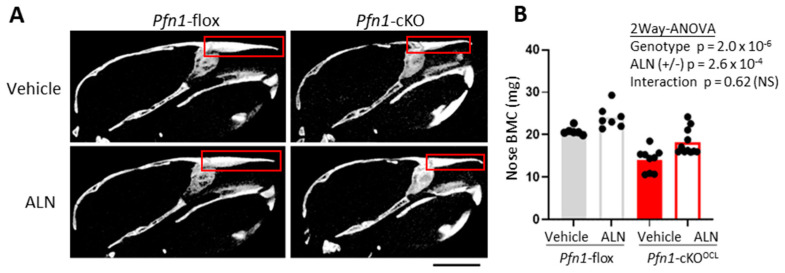
Bone mass analysis of *Pfn1*-cKO^OCL^ L3 nose. (**A**). Representative 2D μCT sagittal images of the skull. The region of interest (ROI) for bone mass analysis was indicated by red boxes. Scale bar: 5 mm (**B**). Nasal bone mineral density (BMC) was plotted and compared among the groups. There were significant differences between the genotypes and ALN treatment, with no significant interactive effect. NS; not significant.

**Table 1 pharmaceuticals-16-01395-t001:** Summary of the trabecular bone mass parameters at distal femur metaphysis.

	*Pfn1*-Flox		*Pfn1*-cKO^OCL^		*p*-Values/Two-Way ANOVA		
	Vehicle (6)	ALN (7)	Vehicle (9)	ALN (11)	Genotype	ALN (+/−)	Interaction
BV/TV (%)	8.1 ± 3.2	25.2 ± 5.8	2.6 ± 1.6	22.2 ± 7.2	0.035	4.8 × 10^−11^	0.50 (NS)
Tb.N (/mm)	0.8 ± 0.6	3.1 ± 0.5	0.2 ± 0.2	2.1 ± 1.6	0.023	3.0 × 10^−6^	0.49 (NS)
Tb.Th (μm)	52.9 ± 2.9	73.9 ± 9.7	31.2 ± 5.3	69.0 ± 8.1	3.2 × 10^−5^	4.0 × 10^−13^	0.0028
Tb.Sp (μm)	722.9 ± 446.7	214.7 ± 66.2	1912.0 ± 1284.6	249.7 ± 116.4	0.033	3.3 × 10^−5^	0.029
Tb.Spac (μm)	774.9 ± 445.5	284.7 ± 69.1	1951.0 ± 1284.4	308.4 ± 116.6	0.036	4.1 × 10^−5^	0.029

NS; not significant.

**Table 2 pharmaceuticals-16-01395-t002:** Summary of the trabecular bone mass parameters at the third vertebral body.

	*Pfn1*-Flox		*Pfn1*-cKO^OCL^		*p*-Values/Two-Way ANOVA		
	Vehicle (6)	ALN (7)	Vehicle (9)	ALN (11)	Genotype	ALN (+/−)	Interaction
BV/TV (%)	7.86 ± 2.36	13.2 ± 5.91	1.46 ± 0.97	7.99 ± 0.69	3.3 × 10^−3^	2.8 × 10^−3^	0.74 (NS)
Tb.N (/mm)	3.87 ± 1.50	4.74 ± 1.42	0.81 ± 0.45	3.33 ± 0.32	4.3 × 10^−4^	6.3 × 10^−3^	0.13 (NS)
Tb.Th (μm)	22.8 ± 0.97	26.7 ± 4.57	17.9 ± 1.41	23.9 ± 0.95	0.01	1.5 × 10^−5^	0.44 (NS)
Tb.Sp (μm)	291.4 ± 103.8	205.9 ± 91.7	1254 ± 506.9	277.6 ± 28.8	2.4 × 10^−4^	5.2 × 10^−4^	7.7 × 10^−4^
Tb.Spac (μm)	314.2 ± 104.3	232.7 ± 88.2	1276 ± 504.6	301.5 ± 29.2	2.4 × 10^−4^	5.2 × 10^−4^	7.2 × 10^−4^

NS; not significant.

## Data Availability

Data is contained within the article and Appendix A.

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
