# Peer review of "Osteolytic Bone Loss and Skeletal Deformities in a Mouse Model for Early-Onset Paget’s Disease of Bone with PFN1 Mutation Are Treatable by Alendronate"

_pharmaceuticals, 2023, doi:10.3390/ph16101395_

Round 1
Reviewer 1 Report
Dear Authors,
Thanks for the interesting research conducted by your team. It is a very interesting therapeutic approach.
Attached I am sending you the document with some suggestions and questions in order to provide us some more information.
Best regards,

Author Response
Reviewer #1
Thank you for your valuable comments. We revised our manuscript as follows.
Comment-1: “Proper and codified symbol for the Profilin 1 mouse gene and protein should be used in the whole manuscript.”
Answer: Thank you very much for your suggestion. We revised all sentences including the description related to human and mouse profilin 1 protein and genes. Human profilin 1 is codified as PFN1, and mouse profilin 1 was codified as Pfn1 according to the general rule. No symbolical abbreviation was applied for profilin 1 protein.
Comment -2: “Keywords - Please change as follows. 1: alendronate; 2: Paget’s disease of bone; 3; Profilin 1; 4: osteoclast; 5: bone deformity.”
Answer: Thank you very much for your suggestion. We changed it as suggested.
Comment -3: “References should be updated.”
Answer: Thank you very much. We updated the references as suggested.
Comment 4: “the section on “early-onset PDB” could be briefly expanded (e.g., increased prevalence of neoplastic degeneration) as also that on “Profilin1” (e.g., suppression of NFKB activity, prevention of the degradation of the phosphatase PTEN).”
Answer: Thank you very much. We revised the paragraphs in the Introduction as follows:
(page 2, lines 47-63)
“PDB is a heterogeneous disorder. Such heterogeneity could be explained partially by multiple different causative or susceptibility genes, and environmental factors (1-3). PDB with mutations in the most frequent causative gene SQSTM1 tends to develop moderate skeletal symptoms in late adulthood (6). In contrast, PDB and its related disorders with mutations other than SQSTM1 tend to develop early and prominent bone deformities. For example, the heterozygous gain of function mutation in TNFRSF11A (RANK) can cause familial symptoms with distinct diagnoses of familial expansile osteolysis and so on (6). Homozygous ZNF687 mutations were found in Italian hereditary cases with earlier and different clinical symptoms in 2016 (1, 7).
More recently, researchers identified the heterozygous loss of function (LOF) mutations in the human profilin 1 (PFN1) gene in some hereditary and sporadic cases of PDB with earlier onset and more prominent clinical symptoms (8-10). The clinical examinations also indicated the increased prevalence of neoplastic degeneration in these cases; by the existence of the cases associated with osteosarcoma (8), or the cases associated with giant cell tumors in addition to the symptoms consistent with PDB (10). Because of such distinctive symptoms and etiology from typical PDB, this disorder was described as “early onset polyostotic PDB”, or “early-onset PDB” in the literature (8-10).”
Suggestion-5: “why the authors decide to test ALN for short-term? Is there a plan to test other agents (e.g., an analogue of denosumab)?.”
Answer: Thank you very much for your asking. We decided to test ALN in this study, because ALN is the most frequently used amino bisphosphonate for osteoporosis in general, and we are familiar with the use of it for animal experiments. Since this was our first trial for testing a therapeutic agent for our mutant line, we dared not test long-term or preventive therapeutic effects in very young mice. Of course, we are thinking about testing different kinds of therapeutic agents with various administration protocols in the future. The following sentences were added or modified in the revised manuscript.
(page 3, lines 93-101)
“Therefore, to evaluate the therapeutic effect of the bisphosphonates in this disorder based on PFN1 mutations, we utilized our mutant mouse line, Pfn1-cKOOCL. The effectiveness of the amino bisphosphonates including zoledronate and alendronate (ALN) is well-known for typical PDB, and zoledronate may be more effective than alendronate. However, since we were unfamiliar with using zoledronate in mouse experiments, we decided to administer the ALN to juvenile mice from 5 to 8 weeks of age, emphasizing the therapeutic effect on bony deformity and bone growth. This timing was chosen because the skeletal deformity became evident in just around 4-5 weeks (15). Short term experiment was chosen because this was the first trial. “
Suggestion-6: “the reason to choose five-week-old mice, that is reported in the section on Animals, should be reported here.”
Answer: Thank you very much. We revised the sentences as stated above answer.
Suggestion-7: “Do the authors think that to report tests for and results of statistical analysis in the panels and/or in the text and/or in the legends is necessary? Some simplifications could improve the fluency of the manuscript.”
Answer: Thank you very much. We tried to reduce the statistical statements in the text and in the figure legend. However, we believe the statements are necessary for presenting the accuracy and appropriateness of our study.
Suggestion-8: “Line 269: do the authors agree to change “skull” with nasal bone?”
Answer: Thank you very much. We changed it to the “nasal bone” here.
Suggestion-9: “Lines 292-295: these sentences are not clear; please reformulate them.”
Answer: Thank you very much. We revised the paragraph as follows.
(page 10, lines 296-307)
“The notion was also supported by histological findings based on TRAP staining. We showed that the ALN treatment not only suppressed the number of TRAP-positive osteoclasts but also improved their abnormal distribution pattern, especially at the endosteum (Fig.2D). It is not surprising to see the osteoclast number decrease by ALN that directly targets the bone-resorbing osteoclasts. However, it was not sure if the altered distribution pattern of the endosteal osteoclasts could also be improved since there has not been shown how the increased osteoclast movements contribute to the altered distribution. The altered distribution was associated with the increased number of osteoclasts in vivo, whereas the osteoclast number was not increased significantly in culture experiments (15). Therefore, we assume that increased movements of osteoclasts in vivo have some influence on osteoclast differentiation or its survival in vivo, which are possible targets of ALN (20, 21). Further investigations are necessary in the future.”
Suggestion-10: “Lines 310-348: please, try to combine and shorten/simplify this part of the text so to make it more fluent. For example, the reasons for the significant difference between the ALN-treated and vehicle-treated mice observed in the femur but not in lumbar vertebra and nasal bone are discussed from line 312 to line 324 and again from line 336 to line 348.”
Answer: Thank you very much. We revised the sentences as follows.
(page 10, lines 318-335)
“Importantly, we saw a significant difference between the ALN-treated and control vehicle-treated mice only in the femur. In the L3 vertebrae and in the nasal bone, we couldn’t prove a significant therapeutic effect of ALN. We suppose a part of this question was related to the fact that the Ctsk-Cre is ectopically expressed in the cambium layer of the perichondrium surrounding the long bone growth plate (22), which corresponds to the regions that we observed remarkable macroscopic changes and therapeutic effect by ALN in our mutant mice (15). An interesting point to this observation was that ALN increased the broadness of the distal quartile of the femur, whereas it decreased the broadness in the Pfn1-cKOOCL mice oppositely. The following are the thinkable interpretations of the observation: in WT, suppression of bone turnover by ALN possibly from the periosteum resulted in increased distal femur width, whereas in Pfn1-cKOOCL mice ALN prevented the enhanced expansion of the width by suppressing the already increased bone resorption from the endosteum. This contrast may correspond to the pattern of the osteoclast distribution in the metaphysis of long bones as indicated in our previous (11) and present study (Fig. 2D). It should be aware that the affected bone turnover at this anatomical site in both ways is known to cause the macroscopic bone deformity named Erlenmeyer flask deformity of the metaphysis. The inhibition as well as enhancement of the bone turnover cause this deformity.
The reasons why only a marginal improvement was observed might be explained in multiple ways. The intervention period of three weeks may have been too short. The timing of the intervention might be too late. In addition, a possible confounding factor was the adverse effect of bisphosphonates on the young growing bone. Caution must be paid regarding their dosage because an overdose of bisphosphonates results in disturbed longitudinal growth of long bones with characteristic Erlenmeyer’s flask deformity at least in mice (23), and maybe in humans (24). Indeed, in our experiments, we saw a slight influence on the length of long bones and vertebrae by ALN, whereas the difference was statistically negligible.”
Suggestion-11: “Lines 351-354: in terms of “prophylactic approach”, why drug should be administered to the “lactating mother”? As the authors report in the section on “Animals”, “the skeletal symptoms begin to be evident” at 5 weeks. Thus, treatment could start in the first month of post-natal life. Please clarify this point.”
Answer: Thank you very much. The skeletal symptoms of our mice become apparent at around 4-5 weeks. However, the changes should have begun much earlier in their life. That is why we are considering the prophylactic approach for them. We revised the sentence as follows.
(page 11, lines 363-365)
“Although we may have to add some arrangements for administrating the drug to the mice before weaning, we may consider it in future studies.”
Suggestion-12: “Line 361-364: which “mild adverse effects” the authors refer to? The experimental system of the authors, rather than to develop “novel agents”, is appropriate for testing available agents (e.g., other more potent aminobisphosphonates) in a short time and “novel agents” in the future. Please check the sentences.”
Answer: Thank you very much. We revised the sentence as follows.
(pages 11-12, lines 372-376)
“Also, the observation of the mild adverse effects of ALN on bone growth indicated that we may have to consider arranging methods of drug administration, selecting the therapeutic agents including other amino bisphosphonates and other available agents, and may need to develop better therapeutic agents whose effects can be modulated controllably at an appropriate level.”
Suggestion-13: “Suggestion-12: “Line 361-364: which “mild adverse effects” the authors refer to? The experimental system of the authors, rather than to develop “novel agents”, is appropriate for testing available agents (e.g., other more potent aminobisphosphonates) in a short time and “novel agents” in the future. Please check the sentences.”
Answer: Thank you very much for your comment. This sentence was revised as answered above.
Suggestion-14: “the legend for Figure S3 is not clear (“……AP projection for male femurs (upper panels) and cross-sectional images at midshaft (bottom) that were analyzed for cortical bone parameters in B, C. Red boxes ……”
Answer: Thank you very much. We revised it as follows.
“A. The length of the femoral shaft as indicated in Fig. 3A was compared on the graph (females). B. The cross-sectional cortical perimeter (Ct.Pm) of the femur at midshaft was plotted on the graph (females). C. Representative mCT images of the male femurs in AP projection (upper panels) and in cross-section at midshaft (bottom) that were analyzed for cortical bone parameters in D-F. Red boxes indicate the ROI for cortical bone analysis. Vertical arrows indicate the length of the proximal and distal halves of the femoral shaft. Horizontal arrows indicate the width (W) of the border between the distal metaphysis and the epiphyses at the growth plate level. Scale bar: 0.5 mm D. The length of the femoral shaft as indicated in C was compared on the graph (males). E. The cross-sectional cortical perimeter of the femurs was plotted on the graph (males). F. Cortical Perimeter (Ct.Pm) normalized by epiphyseal width (W) were plotted on the graph. G. The therapeutic ratio of the above index by ALN was plotted on the graph to compare it between the WT and Pfn1-cKOOCL mice by Student’s t-test (**; p<0.01).”
Suggestion-15: “identify the different groups of mice in the same way.”
Answer: Thank you very much for your comment, but we don’t understand what you mean.
Suggestion-16: “How dose (“0.1 mg/kg body weight”), timing (“twice a week for 3 weeks”) and route of administration (“subcutaneously”) of ALN were established? Was administration based on the human equivalent dose? Were previously published papers considered?”
Answer: Thank you very much. The dose of the ALN was determined according to the human equivalent dose and a lot of previously published papers for mouse experiments.

Reviewer 2 Report
In the manuscript pharmaceuticals-2612786, the authors “evaluated the efficiency of Alendronate (ALN) on a mutant mouse line, recapitulating” early-onset Paget’s Disease of Bone due to PFN1 mutation. To this end, the authors treated “Five-week-old conditional osteoclast-specific Pfn1-deficient mice (Pfn1-cKOOCL) and control littermates (33 females and 22 males) …… with ALN (0.1 mg/kg) or vehicle twice weekly until 8 weeks of age.”. Through histology and 3D micro-CT images, the authors demonstrated improvement of “bone mass at the distal femur, L3 vertebra, and nose” and normalization of osteoclasts and, through the geometric analysis of the bone shape, “a partial recovery from the distal femur deformity”. Based on these findings, the authors concluded that this therapeutic approach in these mice “significantly improved systemic bone loss …… and femoral bone deformity.” and suggest that “preventive treatment of bony deformity in early-onset PDB is feasible”.
Proper and codified symbol for the Profilin 1 mouse gene and protein should be used in the whole manuscript.
Keywords
- Please change as follows: 1: alendronate; 2: Paget’s disease of bone; 3; Profilin 1; 4: osteoclast; 5: bone deformity.
Introduction
- References should be updated (please see below).
- For this reviewer, the section on “early-onset PDB” could be briefly expanded (e.g., increased prevalence of neoplastic degeneration) as also that on “Profilin1” (e.g., suppression of NFKB activity, prevention of the degradation of the phosphatase PTEN).
- “In fact, treatment strategies for early-onset PDB with PFN1 mutations have not been established yet.”. This is true. However, in humans, the response to aminobisphosphonates, including zoledronic acid (ZOL), has been reported to be reduced and, in most patients, multiple infusions were required to control bone pain and achieve biochemical remission (doi: 10.1210/clinem/dgaa252; doi: 10.1002/jbmr.4275). In addition, ALN is known to be less potent than ZOL. Thus, why the authors decide to test ALN for short-term? Is there a plan to test other agents (e.g., an analogue of denosumab)?
- For this reviewer, the reason to choose five-week-old mice, that is reported in the section on Animals, should be reported here.
Results
- They are overall clearly presented. Do the authors think that to report tests for and results of statistical analysis in the panels and/or in the text and/or in the legends is necessary? Some simplifications could improve the fluency of the manuscript.
Discussion
- Line 269: do the authors agree to change “skull” with nasal bone?”.
- Lines 292-295: these sentences are not clear; please reformulate them.
- Lines 310-348: please, try to combine and shorten/simplify this part of the text so to make it more fluent. For example, the reasons for the significant difference between the ALN-treated and vehicle-treated mice observed in the femur but not in lumbar vertebra and nasal bone are discussed from line 312 to line 324 and again from line 336 to line 348.
- Lines 351-354: in terms of “prophylactic approach”, why drug should be administered to the “lactating mother”? As the authors report in the section on “Animals”, “the skeletal symptoms begin to be evident” at 5 weeks. Thus, treatment could start in the first month of post-natal life. Please clarify this point.
- Line 361-364: which “mild adverse effects” the authors refer to? The experimental system of the authors, rather than to develop “novel agents”, is appropriate for testing available agents (e.g., other more potent aminobisphosphonates) in a short time and “novel agents” in the future. Please check the sentences.
- Line 374: is “color” for collar?
References
They should be updated. Please see for example doi: 10.3389/fgene.2023.1131182 and doi: 10.3389/fcell.2022.932065.
Figures and Legends
- Please c the figures and their legends. For example, in Figure 1 “LS” should be clarified, in Figure 1 and in Figure 2 the bar could be moved within 3D microCT image and histological image in panel B and D respectively, in Figure 4 (panel B) and in Figure 5S (panels B and E) “NS” can be removed, in Figure S2 Pfn1-cKOOCL should be centered and the legend for Figure S3 is not clear (“……AP projection for male femurs (upper panels) and cross-sectional images at midshaft (bottom) that were analyzed for cortical bone parameters in B, C. Red boxes ……”). In addition, this reviewer suggests to identify the different groups of mice in the same way.
Animals
How dose (“0.1 mg/kg body weight”), timing (“twice a week for 3 weeks”) and route of administration (“subcutaneously”) of ALN were established? Was administration based on the human equivalent dose? Were previously published papers considered?
Moderate editing of English language required.
Author Response
Thank you for your comment. We revised our manuscript as follows.
Comment 1.
“Lines 82-85: This sentence is not clear. It should be clarified.”
Answer: Thank you very much for your question.
As you mentioned, .
(Page 2, lines 85-882)
“Instead, from the aspect of actin-filament reorganization and cellular movements, we have shown that our mice had osteoclasts with increased migratory activity, increased cellular processes, and increased bone resorption activity which were mutually correlated in culture experiments (15, 16). “
Comment 2.
“The authors should present a rationale for their choice of ALN among all bisphosphonates.”
Answer: Thank you for your very important comment. We revised our manuscript as follows.
(Page 3, line 94-99)
“The effectiveness of the amino bisphosphonates including zoledronate and alendronate (ALN) is well-known for typical PDB, and zoledronate may be more effective than alendronate. However, since we were not familiar with the use of zoledronate in mouse experiments, we decided to administer the ALN to juvenile mice from 5 to 8 weeks of age, emphasizing the therapeutic effect on bony deformity and bone growth. “
Comment 3.
“The authors should provide disclosure of all abbreviations in the footnotes as well as on first use in the text.”
Answer: Thank you very much. We provided the list of all abbreviations in the manuscript, as follows.
(Page lines )
“Therefore,
Comment 4.
“Line 243: The difference between BMD and BMC should be indicated.”
Answer: Thank you for your suggestion. We revised the following sentences in the text.
(Page 9, lines 248-254)
“To this end, we focused on the nasal bone on the mCT images in the midsagittal plane and compared its BMC (mg). We haven’t chosen to compare the vBMD (g/cm3) for our evaluation, because the Pfn1-cKOOCL mice had apparently smaller nasal bone compared to the WT (15), and this was supposed to be the result of fragile bone fracture caused by daily mechanical stress. Thus, we compared the nasal bone BMC and showed that in the Pfn1-cKOOCL mice was significantly improved by ALN treatment (Fig. 6A, B).”
Comment 5.
“Discussion should not contain references on Figures or Tables. This should be corrected.”
Answer: Thank you very much. We quit to quote them that were presented in Results.
Comment 6.
“Study limitations and conclusions are missing. This should be corrected.”
Answer: We understood that our descriptions related to the limitations and conclusions in the discussion were not clear to you in our previous manuscript. We now provide it more clearly in the text, as follows.
(Page 12, lines 379-396)
“In summary, this study indicated that standard bisphosphonate therapy can treat bone loss in the early onset PDB based on Pfn1 mutation in a mouse model system. In addition, our study also indicated that the skeletal deformity might be prevented or recovered by ALN at least partially even after disease onset. The limitations of our study were that the therapeutic effects of ALN on bony deformity were overall marginal, and different administration protocols were not tested in this study. Further investigations applying different administration protocols and therapeutic agents should be conducted in the future which may lead to establishing a better sophisticated therapeutic protocol to treat young patients with early onset PDB with skeletal deformities. Biologically, our results support the hypothesis that the bone phenotype of Pfn1-cKOOCL mice is caused mainly by osteoclast-specific dysfunction. However, still, there is still a possibility that Pfn1 mutation in mesenchymal progenitor cells in the cambium layer that gives rise to bone collar osteoblasts or chondrocytes may oppositely function to resist deformity.
- Conclusion
Our study indicated that skeletal phenotypes of the Pfn1-cKOOCL mice including the bony deformity were improved by short-term ALN administration from 5 to 8 weeks of the mice.”
Comment 7.
“References should be arranged in accordance with Journal’s requirements..”
Answer: Thank you very much. We checked it again, according to your suggestion.

Reviewer 3 Report
The authors studied consequences of ALN treatment of mice with PFN1mutation as a model of early-onset Paget’s disease of bone. They noticed that ALN improved osteolytic bone loss and femoral deformity in mice.
Comments
1. Lines 82-85: This sentence is not clear. It should be clarified.
2. 2. The authors should present a rationale for their choice of ALN among all bisphosphonates.
3. All Tables and Figures: The authors should provide disclosure of all abbreviations in the footnotes as well as on first use in the text.
4. Line 243: The difference between BMD and BMC should be indicated.
5. Discussion should not contain references on Figures or Tables. This should be corrected.
6. Study limitations and conclusions are missing. This should be corrected.
7. References should be arranged in accordance with Journal’s requirements.
All the typos should be corrected.
Author Response
Thank you for your comment. We revised our manuscript as follows.
Comment 1.
“Line 93: Have you checked for adverse effects concerning the craniofacial development?
Were these effects observed only in appendicular bones?”
Answer: Thank you very much for your question. We tried to check if some adverse or therapeutic effects on craniofacial development by ALN in this study. However as indicated by the analysis on the bony deformity in the spine and the skull, it was hard to point out such effects. This was supposed to be due to a lack of appropriate methods of comparisons in part. Therefore, we would like to pursue it in the next studies.
Comment 2.
“Line 96-97: Emphasize in which bones these effects are observed.”
Answer: Thank you very much for your question. We revised the title as follows.
“Administration of the ALN improved the decreased trabecular bone mass in the distal femur of the Pfn1-cKOOCL mice.
Comment 3.
“Figure 1A: Please precise the the mice age. In the abstract you talk about the mice age in weeks and here they are in days.”
Answer: Thank you very much for your pointing out this. We revised it as suggested.
Comment 4.
“Line 116: Watch out for the animals' age.”
Answer: Thank you very much for your question. The administration period was 3 weeks as written here. The administration was done from 5 to 8 weeks of mouse age.
Comment 5.
“Table 1: Provide these data in graphics. It is easier to aprehend and it is visually more attractive.”
Answer: Thank you very much for your suggestion. However, since we have already presented the graphs for BV/TV in Figure 1, we didn't add new figures for other different parameters representing bone mass. We believe the readers can imagine what they are without further graph presentation. The message does not change.
Comment 6.
“Line 150: cells. TRAP is an osteoclast marker.”
Answer: Thank you very much for your suggestion. We regarded the TRAP-positive cells with nucleus number ≧ 3 as osteoclasts.
Comment 7.
“Line 285: Please just explain the rationale for choosing this specific BPS instead of the others.”
Answer: Thank you very much for your question. We decided to test ALN in this study, because ALN is the most frequently used amino bisphosphonate for osteoporosis in general, and we are familiar with the use of it for animal experiments. Since this was our first trial for testing a therapeutic agent for our mutant line, we dared not test long-term or preventive therapeutic effects in very young mice. We revised the sentences at the end of the Introduction (lines 95-103) as follows.
“Therefore, to evaluate the therapeutic effect of the bisphosphonates in this disorder based on PFN1 mutations, we utilized our mutant mouse line, Pfn1-cKOOCL. The effectiveness of the amino bisphosphonates including zoledronate and alendronate (ALN) is well-known for typical PDB, and zoledronate may be more effective than alendronate. However, since we were unfamiliar with using zoledronate in mouse experiments, we decided to administer the ALN to juvenile mice from 5 to 8 weeks of age, emphasizing the therapeutic effect on bony deformity and bone growth. This timing was chosen because the skeletal deformity became evident in just around 4-5 weeks (15). Short term experiment was chosen because this was the first trial. “
Comment 8.
“Line 336: Was the tested time lapse for drug administration randomly chosen or did you go through a pilot study for your choices?”
Answer: Thank you very much for your question. The drug administration protocol was chosen based on the literature. The age of the mice was chosen based on the phenotype of our mice as described above.

Round 2
Reviewer 2 Report
Dear Authors,
this reviewer thanks you for the detailed point by point response to the comments/suggestions.
Regarding "Suggestion-15", I'm really sorry for the misunderstanding. For this reviewer, the graphs regarding the therapeutic ratio (two columns) could be presented in different colors because they are presented as for Pfn1-flox-vehicle and Pfn1-flox-ALN.
Best regards
Minor editing of English language required.